# Highlighting the Crude Oil Bioremediation Potential of Marine Fungi Isolated from the Port of Oran (Algeria)

**Ahlem Maamar [1], Marie-Elisabeth Lucchesi [2], Stella Debaets [2], Nicolas Nguyen van Long [3], Maxence Quemener [2], Emmanuel Coton [2], Mohammed Bouderbala [1], Gaëtan Burgaud [2,\*] and Amaria Matallah-Boutiba [1]**

[1] Laboratoire Réseau de Surveillance Environnementale (LRSE), Department of Biology, Faculty of Life and Nature, University of Oran, Oran 31000, Algeria; maamarahlem@yahoo.fr (A.M.); bouderbalamoh@yahoo.fr (M.B.); amariamatallah@hotmail.com (A.M.-B.)

[2] Laboratoire Universitaire de Biodiversité et Ecologie Microbienne, Université de Brest, F-29280 Plouzané, France; lucchesi@univ-brest.fr (M.-E.L.); stella.debaets@univ-brest.fr (S.D.); maxence.quemener@univ-brest.fr (M.Q.); emmanuel.coton@univ-brest.fr (E.C.)

[3] UMT19.03.01 ALTER'iX, Food Quality & Safety Unit, ADRIA Food Technology Institute, 29000 Quimper, France; Nicolas.NGUYENVANLONG@adria.tm.fr

\* Correspondence: gaetan.burgaud@univ-brest.fr

**Abstract:** While over hundreds of terrestrial fungal genera have been shown to play important roles in the biodegradation of hydrocarbons, few studies have so far focused on the fungal bioremediation potential of petroleum in the marine environment. In this study, the culturable fungal communities occurring in the port of Oran in Algeria, considered here as a chronically-contaminated site, have been mainly analyzed in terms of species richness. A collection of 84 filamentous fungi has been established from seawater samples and then the fungi were screened for their ability to utilize and degrade crude oil. A total of 12 isolates were able to utilize crude oil as a unique carbon source, from which 4 were defined as the most promising biodegrading isolates based on a screening test using 2,6-dichlorophenol indophenol as a proxy to highlight their ability to metabolize crude oil. The biosurfactant production capability was also tested and, interestingly, the oil spreading and drop-collapse tests highlighted that the 4 most promising isolates were also those able to produce the highest quantity of biosurfactants. The results generated in this study demonstrate that the most promising fungal isolates, namely *Penicillium polonicum* AMF16, *P. chrysogenum* AMF47 and 2 isolates (AMF40 and AMF74) affiliated to *P. cyclopium*, appear to be interesting candidates for bioremediation of crude oil pollution in the marine environment within the frame of bioaugmentation or biostimulation processes.

**Keywords:** marine fungi; crude oil; biodegradation; DCPIP; biosurfactant

## 1. Introduction

Oil activity is growing at a fast rate leading to the development of massive offshore and onshore infrastructures for exploration, drilling, transportation, storage, processing and delivery, with many critical steps that can cause serious environmental and health threats due to chronic or accidental crude oil spillage [1–5]. Crude oil is a natural heterogeneous mixture composed of hydrocarbon deposits and other organic materials, including among others alkanes, cycloalkanes and aromatic hydrocarbons with various hazardous, toxic and carcinogenic potential [6,7]. Different techniques have been developed for restoration of oil-contaminated habitats, including physical, chemical and biological treatments [8].

While the generally used physicochemical approach seems neither effective nor cost-efficient [9], bioremediation appears as an environmentally friendly, efficient and economical alternative [10,11].

Fungi represent an important group of microorganisms occurring in a wide range of marine habitats from coastal waters (e.g., [12–14]) to the deep biosphere (e.g., [15–17]) and associated to various marine substrates such as intertidal sediments [18,19], drifted wood [20], algae [21] and animals such as fish, prawns or even mussels [22,23]. Fungal occurrence appears well correlated with organic matter, suggesting important roles as recyclers of complex polymers in the ocean such as polysaccharides [24], lignocellulose [25] and even hydrocarbon-based polymers such as microplastics [26,27] and hydrocarbons.

Hydrocarbons represent a rich source of energy and carbon for microorganisms able to degrade them, named hydrocarbonoclasts. Almost 100 bacterial genera as well as numerous archaeal genera capable of degrading selected hydrocarbons have been identified [28,29] while over 100 fungal genera are known to play non-trivial roles in the biodegradation of hydrocarbons [30]. Fungi affiliated to the *Cladosporium* and *Aspergillus* genera are among those known to participate in aliphatic hydrocarbon degradation while *Cunninghamella*, *Penicillium*, *Fusarium*, *Aspergillus* and *Mucor* representatives have been shown to take part in the degradation of more recalcitrant aromatic hydrocarbons [31]. Microbial degradation of hydrocarbons involves complex enzymatic activities such as those of monooxygenases, dioxygenases, hydroxylases, dehydrogenases, oxidoreductases, etc. [32,33]. While the pathways of hydrocarbon degradation by bacteria have been well studied, knowledge of enzymatic mechanisms and associated genetic pathways of hydrocarbon degradation in fungi appears much more limited [34]. Fungi are thought to play a critical role in facilitating the degradation of recalcitrant hydrocarbons by secreting extracellular enzymes transforming these compounds into intermediates of lower environmental toxicity and increasing further decomposition by bacteria [35]. This idea is supported by culture-based studies that have detected increased degradation of poly-aromatic hydrocarbons when different fungi were added [36–38].

Studies in the field of bioremediation by fungi have so far focused mainly on terrestrial environments [34,39–41].Very few studies have focused on fungal bioremediation of petroleum in the marine environment [1,2,42–45]. Effective biodegradation of crude oil by marine fungi was demonstrated through quantification of changes in the total mass of crude oil over time [45]. Fungi isolated from hydrocarbon-contaminated beaches in the Gulf of Mexico were also reported as able to degrade n-alkanes and polycyclic aromatic hydrocarbons [44]. Some fungi may also play a critical and complementary role in facilitating the bio-availability of hydrocarbons to other microbial communities (i.e., to other fungi and/or bacteria) by synthesizing biosurfactants. The poor bioavailability of hydrocarbon components is considered a major limiting factor in the hydrocarbon remediation process [46]. In this context, biosurfactants act as surface-active amphiphilic compounds with a hydrophobic and hydrophilic moiety, interacting with phase boundaries in a heterogeneous system to solubilize organic compounds [47]. The entire phenomenon enhances the bioavailability through better solubilization of hydrocarbons in water or water in hydrocarbons [48]. Biosurfactants play major roles in various fields such as bioremediation, biodegradation, and oil recovery. Chemical surfactants exist (carboxylates, sulfonates, sulfates) [49], but biosurfactants have several advantages such as lower toxicity and higher biodegradability [50]. While the a majority of described biosurfactants are of bacterial origin, with producers affiliated to the *Pseudomonas*, *Acinetobacter*, and *Bacillus* genera [51], the production of biosurfactants by yeasts and filamentous fungi has recently been of growing interest. Fungal biosurfactant producers are affiliated to the *Candida*, *Pseudozyma* or *Rhodotorula* genera for yeasts [47,52,53] and to the *Cunninghamella*, *Fusarium*, *Phoma*, *Cladophialophora*, *Exophiala*, *Aspergillus* and *Penicillium* genera for filamentous fungi [54,55]. A recent study has highlighted that some microbial strains could even synthesize biosurfactants by utilizing crude oil components as carbon sources, thereby leading to an improved degradation of hydrocarbons [56].

The main objectives of this study were: (i) to isolate and identify culturable fungal communities occurring in the port of Oran in Algeria as a chronically hydrocarbon-contaminated site; (ii) to

screen the ability of isolates to utilize crude oil as unique carbon source and (iii) to check their ability to produce biosurfactants. The aim here was to gain insights into the evaluation of fungal bioremediation as a promising, efficient and ecological alternative to the physicochemical processes of hydrocarbon-contaminated site restoration.

## 2. Material and Methods

### 2.1. Sampling and Isolation

Seawater samples were randomly collected in the port of Oran, Algeria (35°42′31.07′′ N; 0°38′26.76′′ W) in 2017 from January to September using sterile glass vials (Figure 1). The seawater samples were collected just below the surface (~20 cm depth). Sea surface samples were directly stored at 4 °C after sampling and treated for fungal isolation in less than 4 h. Seawater samples (1 mL) were spread onto Sabouraud agar plates prepared with sterilized seawater from the sampling site to mimic in situ conditions (peptone 10 g, glucose 40 g, agar 15 g, natural seawater 1 L) and supplemented with chloramphenicol (50 mg/L) to inhibit bacterial growth. All Petri dishes were incubated at 24 °C for a period of 7 days. Fungal colonies were then isolated in pure culture for further taxonomic identification.

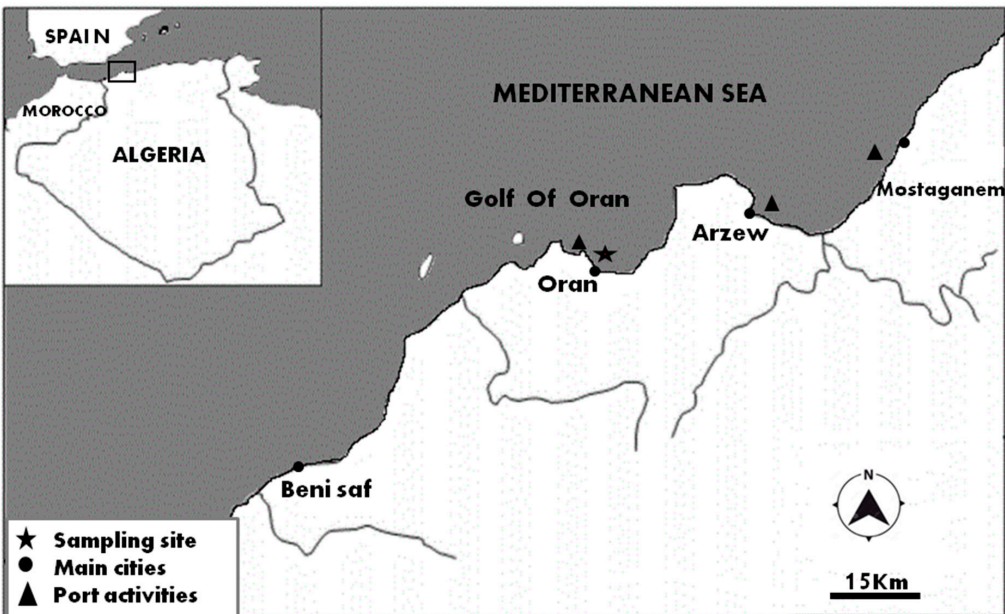

**Figure 1.** Localization of the chronically-contaminated sampling site.

### 2.2. Identification of Fungal Isolates

DNA extraction from fungal isolates was performed using a Fast DNA Spin Kit (MP Biomedicals) following the manufacturer's instructions. DNA quality and quantity were checked using a NanoDrop 1000 spectrophotometer (Thermo Scientific). The extracted DNA was then standardized at 5 ng/μL and stored at −20 °C before amplification of taxonomic markers. A morphological dereplication was performed to cluster similar isolates, then different representatives of each morphotype cluster were selected for genetic identification based on internal transcribed spacer (ITS) region sequencing using ITS4 (5′-TCCTCCGCTTATTGATGATATGC-3′) and ITS5 (5′-GGAAGTAAAAGTCGTAACAAG-3′) primers [16,57]. Based on the taxonomic identification, complementary genetic markers were sequenced to refine assignation. For *Penicillium* and *Aspergillus* isolates, the ß-tubulin gene sequence was amplified using BT2a (5′-GGTAACCAAATCGGTGCTGCTTTC-3′) and BT2b (5′-ACCCTCAGTGTAGTGACCCTTGGC-3′) primers, while for *Cladosporium* isolates, the actin gene sequence was amplified using the Act 512F (5′-ATGTGCAAGGCCGGTTTCGC-3′) and Act 783R (5′-TACGAGTCCTTCTGGCCCAT-3′) primers. All PCRs were performed in reaction volumes of 25 μL

containing 1X PCR buffer, 0.4 mM dNTPs (Promega), 0.36 μM of each primer (Sigma), 2 mM MgCl2, 1 U of Taq-DNA polymerase (Promega) and 1 μL of genomic DNA. The amplification cycle included a denaturation step at 94 °C for 2 min, followed by 30 cycles of 30 s at 94 °C, 30 s at 50 °C for ITS, 61 °C for ß-tubulin and 58 °C for actin, and 2 min at 72 °C followed by a final elongation step of 7 min at 72 °C. A negative control in which DNA was replaced by sterile water was included. The PCR products were analyzed by electrophoresis in 1.4% agarose gel (Promega) in 1X TBE buffer at 85 V for 1 h and stained with GelRed (Thermo Fischer Scientific). The amplicons were then visualized under UV and the image files were generated using a Quantum ST4 system (Vilbert Lourmat). All PCR products obtained were sequenced on both strands by Eurofins Scientific (Germany) using the same pairs of primers used to amplify the targeted regions. Sequences were then assembled using Sequencher 5.4 (Gene Codes) and analyzed using BLASTN (NCBI) to affiliate each sequence to the species level. All sequences can be accessed on NCBI using the following accession numbers: (i) from MN794435 to MN794486 for ITS sequences, (ii) from MT427896 to MT427900 for actin sequences and (iii) from MT427868 to MT427895 for B-tubulin sequences.

## 2.3. Screening of Fungal Growth on Crude Oil

All isolated fungi were tested for their ability to use Arabian Light crude oil as a unique carbon source. For each fungal strain, a spore suspension was prepared in sterile water supplemented with tween 80 (0.1%) and standardized at $10^5$ spores/mL. A volume of 10 μL was then used to inoculate each well of a 12-well microplate containing triplicates of 4 different media: Czapek–Dox agar medium (3 g/L $NaNO_3$, 1 g/L $K_2HPO_4$, 0.5 g/L $MgSO_4$, 0.5 g/L KCl, 0.01 g/L $FeSO_4$, 18 g/L agar and 3% sea salts) supplemented with 3 different concentrations of Arabian Light crude oil (0.1%, 1%, 5%) as a unique carbon source. A control in which crude oil was replaced by sucrose (30 g/L) was also included as the 4th medium. The microplates were then incubated at 24 °C. Radial growth was analyzed by measuring the diameter of the thallus at different incubation time points (every 48 h over 6 days). The thallus diameter extension as a function of incubation time was modelled with a two-phase primary model with latency [58,59] as followed:

$$r(t) = \begin{cases} r_0, \ t \leq \lambda \\ r_0 + \mu \cdot (t - \lambda), \ t > \lambda \end{cases}$$

where r(t) is the thallus radius (mm) at the time of incubation t (days), $r_0$ is the thallus measured at time = 0 days, $\lambda$ is the latency before growth (days) and μ represents the radial growth rate (mm·day$^{-1}$). In the present study, $r_0$ was fixed at 0 mm. The model was fitted by minimizing the sum of squares of the residuals with the *lsqcurvefit* function of Matlab (R2019a, The Mathworks Inc., Natick, MA, USA). The confidence intervals (95%) for the estimated primary parameters were computed by linear approximation with the *nlrparci* function of Matlab. The fitting performances of the models were assessed on the basis of determination coefficient (r2) and root mean square error (RMSE).

## 2.4. Crude Oil Degradation Assay

Twelve isolates were selected from the previous experiment and tested for their ability to degrade crude oil based on protocols from Varjani and Upasani (2013) [60] and Bovio et al. (2017) [45], but modified here to be used in agar-based cultures. A suspension of spores for each strain was prepared as previously explained and used to inoculate 12-well microplates with the same conditions except that (i) only 1% crude oil condition was tested and (ii) 2,6-dichlorophenol indophenol (DCPIP) was supplemented in each well at 1 g/L. The discoloration diameter of DCPIP (around fungal colonies) from blue to colorless, as a proxy for crude-oil degradation capability, was measured for each well, including the controls (with crude oil replaced by sucrose), every 48 h during 6 days of incubation at 24 °C.

## 2.5. Biosurfactant Production

Twelve fungal isolates were grown in 100 mL of a specific medium (20 g/L rapeseed oil, 5 g/L glucose, 5 g/L yeast extract), previously designed to stimulate the production of biosurfactants [61],

using a concentration of $2.5 \times 10^6$ spores/mL of a spore suspension for inoculation. Flasks were then incubated at 28 °C for 96 h in agitated conditions (150 rpm). After incubation, cultures were centrifuged at 9000 rpm for 15 min in order to collect and filter the supernatant using sterile filtration units and membranes of 0.45 μm (Millipore®). Detection of biosurfactant-producing fungi was assessed using drop-collapsing and oil-spreading tests as sensitive and rapid methods. The drop-collapsing test consisted of pipetting 100 μL of supernatant onto the hydrophobic surface of a parafilm sealing film. After 1 min the diameter of each drop was measured and compared to the diameter of drops of: (i) distilled water, (ii) culture medium (as negative controls) and (iii) 10% sodium lauryl sulfate (as a positive control) following Bodour and Maier (1998) [62]. The oil-spreading test consisted of pipetting 15 μL of supernatant from each isolate in a Petri dish filled with 25 mL of distilled water and 15 μL of Arabian Light crude oil. The presence of biosurfactants in a supernatant lead to the formation of a halo which can be measured and compared to the positive and negative controls [63].

## 3. Results

### 3.1. Diversity of Culturable Fungal Communities in the Port of Oran

A total of 84 filamentous fungal isolates belonging to 30 different taxa were obtained and genetically identified here through the sequencing of several genetic markers. All isolates belong to the Ascomycota phylum and were clustered into eight different genera (*Penicillium, Aspergillus, Cladosporium, Alternaria, Phialemonium, Acremonium, Chaetomium, Scedosporium*) and one order (*Pleosporales*). The culturable diversity is here dominated by the genera *Aspergillus* (11 species), *Penicillium* (nine species) and *Cladosporium* (five species), representing 36.7%, 30% and 16.7%, respectively, of the whole diversity, in terms of species richness. In terms of abundance, *Aspergillus flavus* appears as the most abundant species with seven isolates, followed by *Aspergillus pseudoglaucus* (six isolates) and *Cladosporium cladosporioides* (six isolates). The fungal diversity was compared to the literature in order to highlight the fungal species that were retrieved for the first time in the marine environment (Table 1) and revealed a total of 12 taxa, including five species affiliated to the *Aspergillus* genus (*A. ruber, A. cibarius, A. pseudoglaucus, A. tritici* and *A. ochraceopetaliformis*), three to the *Cladosporium* genus (*C. aggregatocicatricatum, C. limoniforme* and *C. dominicanum*) and two species affiliated to the *Penicillium* genus (*P. mononematosum* and *P. cyclopium*).

**Table 1.** List of fungal isolates and their taxonomic affiliation depending on several genetic markers. Species highlighted for the first time in the marine environment are in bold and those highlighted for the first time in seawater are in bold with a star.

| Sample Type / Fungal Species | Number of Isolates | Seawater/Sediments | Fish/Marine Invertebrates/Algae/Marine Plant (*Posidonia oceanica*) |
|---|---|---|---|
| *Acremonium* sp. | 1 (AMF100) | (53), (52) | (54) |
| *Alternaria alternata* | 2 (AMF7, AMF24) | (3), (51) | (16), (30), (31), (32) |
| **Aspergillus cibarius** | 5 (AMF19, AMF93, AMF26, AMF45, AMF105) | - | - |
| *Aspergillus flavus* | 7 (AMF29, AMF22, AMF20, AMF46, AMF57 AMF80, AMF90) | (5), (7), (8), (51) | (8), (14), (27) |
| **Aspergillus ochraceopetaliformis** | 1 (AMF68) | - | - |
| *Aspergillus ostianus* | 3 (AMF12, AMF8, AMF3) | - | (47) |
| **Aspergillus pseudoglaucus** | 6 (AMF23, AMF21, AMF82, AMF83, AMF94, AMF43) | (56) | (55) |

Table 1. *Cont.*

| Sample Type / Fungal Species | Number of Isolates | Seawater/Sediments | Fish/Marine Invertebrates/Algae/Marine Plant (*Posidonia oceanica*) |
|---|---|---|---|
| *Aspergillus ruber* | **5** (AMF17, AMF18, AMF81, AMF48, AMF54,) | - | - |
| *Aspergillus sydowii* | **2** (AMF15 AMF77) | (5), (7), (39), (56) | (11), (24) |
| *Aspergillus tritici* | **3** (AMF44, AMF78, AMF106) | - | - |
| *Aspergillus versicolor* | **4** (AMF75, AMF97, AMF92, AMF104) | (3), (7), (51) | (11), (14), (16) |
| *Aspergillus wentii* * | **1** (AMF61) | - | (28) |
| *Aspergillus westerdijkiae* | **2** (AMF42, AMF60) | (3), (56) | - |
| *Chaetomium globosum** | **1** (AMF31) | - | (33), (34), (35), (36) |
| *Cladosporium aggregatocicatricatum* | **1** (AMF65) | - | - |
| *Cladosporium cladosporioides* | **6** (AMF91, AMF36, AMF87, AMF95, AMF63, AMF89) | (3), (40), (41) | (37), (38), (42), (43), (44), (45), (46) |
| *Cladosporium dominicanum* | **3** (AMF88, AMF96, AMF51 | - | - |
| *Cladosporium herbarum** | **1** (AMF85) | - | (29) (16) |
| *Cladosporium limoniforme* | **1** (AMF35) | - | - |
| *Penicillium brevicompactum* | **2** (AMF34, AMF86) | (3), (6), (57) | (20), (11), (14), (16), (21), (22), (12) (23), (19), (24) |
| *Penicillium chrysogenum* | **4** (AMF4, AMF33, AMF40, AMF76) | (1), (2), (3), (4), (5), (40), (58) | (9), (10), (11), (12), (13), (14), (15), (16), (17) |
| *Penicillium citreonigrum* | **3** (AMF39, AMF102, AMF103) | (3) | (15), (25), (26) |
| *Penicillium cyclopium* | **3** (AMF47, AMF73 AMF74) | - | - |
| *Penicillium glabrum** | **1** (AMF56) | - | (11) |
| *Penicillium mononematosum* | **2** (AMF27, AMF79) | - | - |
| *Penicillium olsonii* | **1** (AMF72) | (3) | (14) |
| *Penicillium polonicum** | **1** (AMF16) | (49), (50), (51) | (14), (18), (48) |
| *Penicillium steckii* | **2** (AMF25, AMF66) | (6) | (11), (19) |
| *Phialemonium inflatum* | **1** (AMF9) | - | - |
| *Pleosporales* sp. | **6** (AMF10, AMF6, AMF14, AMF11, AMF37, AMF38) | n.d | n.d |
| *Scedosporium sp.* | **3** (AMF67, AMF71, AMF28) | (59) | (60) |

(1) Wang et al. (2018) [64], (2) Luo et al. (2014) [65], (3) Bovio et al. (2017) [45], (4) Elshafie et al. (2007) [66], (5) Alwakeel (2017) [67], (6) Marchese et al. (2016) [68], (7) Barnes et al. (2018) [69], (8) Zuluaga-Montero et al. (2010) [70], (9) Visamsetti et al. (2016) [71], (10) Gao et al. (2011) [72], (11) Paz et al. (2010) [73], (12) Passarini et al. (2013) [74], (13) Ding et al. (2011) [75], (14) Pivkin et al. (2006) [76], (15) Wiese et al. (2011) [77], (16) Panno et al. (2013) [78], (17) Bovio et al. (2019) [79], (18) Neethu et al. (2018) [80], (19) Garzoli et al. (2018) [81], (20) Gnavi et al. (2017) [21], (21) Lopez-Legenthil et al. (2015) [82], (22) Mabrouk et al. (2010) [83], (23) Pivkin et al. (2006) [76], (24) Godinho et al. (2019) [84], (25) Jones et al. (2015) [85], (26) Raghukumar and Ravindran (2012) [86], (27) Lin et al. (2008) [87], (28) Sun et al. (2012) [88], (29) Shigemori et al. (2004) [89], (30) Gao et al. (2009) [90], (31) Shaaban et al. (2012) [91], (32) Shi et al. (2017) [92], (33) Wang et al. (2006) [93], (34) Cui et al. (2010) [94], (35) Yamada et al. (2009) [95], (36) Muroga et al. (2010) [96], (37) Henríquez et al. (2014) [97], (38) Rozas et al. (2011) [98], (39) Damare et al. (2006) [99], (40) Pindi et al. (2012) [100], (41) Shao and Sun, (2007) [101], (42) Manriquez et al. (2009) [102], (43) San-Martin et al. (2005) [103], (44) Sayed et al. (2016) [104], (45) Bonugli-Santos et al. (2010) [105], (46) Hulikere et Joshi, (2019) [106], (47) Namikoshi et al. (2003) [107], (48) Xin et al. (2007) [108], (49) Song et al. (2012) [109], (50) Yu et al. (2010) [110], (51) Abo-Kadoum et al. (2013) [111], (52) Chung et al. (2019) [112], (53) Rédou et al. (2015) [16], (54) Zuccaro et al. (2004) [113], (55) Smetanina et al. (2007) [114], (56) Lee et al. (2016) [115], (57) Velez et al. (2020) [116], (58) Nagano et al. (2017) [117], (59) Li et al. (2020) [118], (60) Raghukumar (2017) [119].

### 3.2. Ability to Grow on Crude Oil

All fungal isolates were tested for their ability to grow in the presence of Arabian Light crude oil as a unique carbon source. Among the 84 tested fungal isolates, 12 were able to grow at one or two concentrations of crude oil, i.e., 1% and 5% (Figure 2). No fungal isolates were able to grow at 0.1% of crude oil, which can also be considered as a negative control here, highlighting the incapacity of fungi to grow using agar as a carbon source and thus emphasizing the idea that these 12 fungal isolates truly utilize crude oil as a unique carbon source.

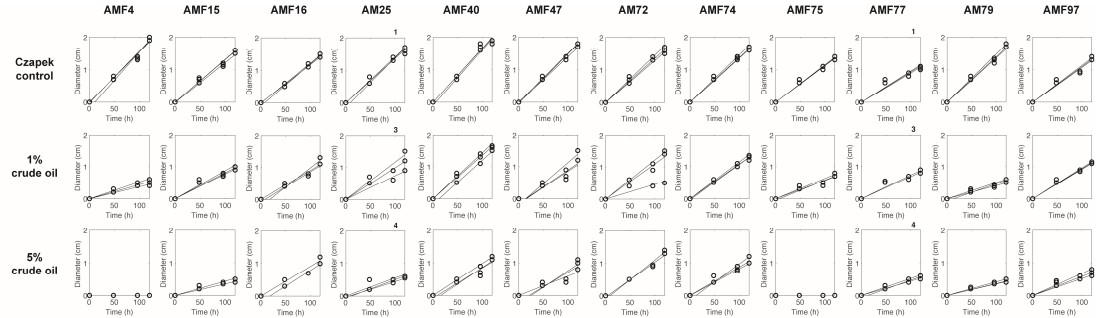

**Figure 2. Growth capabilities of the 12 most promising fungal isolates with 1% and 5% crude oil as unique carbon source.** AMF4: *Penicillium chrysogenum*; AMF15: *Aspergillus sydowii*; AMF16: *Penicillium polonicum*; AMF25: *Penicillium steckii*; AMF40: *Penicillium cyclopium*; AMF47: *Penicillium cyclopium*; AMF72: *Penicillium olsonii*; AMF74: *Penicillium cyclopium*; AMF75: *Aspergillus versicolor*; AMF77: *Aspergillus sydowii*; AMF79: *Penicillium mononematosum* and AMF97: *Aspergillus versicolor*. The growth capacity of the isolates with 0.1% crude oil was also tested but no growth was observed.

Growth rates were determined for each positive isolate and each condition. The highest growth rates were observed (i) for *Aspergillus sydowii* AMF15, *Penicillium polonicum* AMF16 and *Aspergillus versicolor* AMF97 at 1% of crude oil, with values reaching 0.0081, 0.0101 and 0.0094 mm/h, respectively, and (ii) for *Penicillium chrysogenum* AMF40, *Penicillium cyclopium* AMF47, *Penicillium olsonii* AMF72 and *P. cyclopium* AMF74 at 1% and 5% crude oil, with values reaching 0.0138 and 0.0097, 0.0112 and 0.0083, 0.0092 and 0.0114 mm/h, and 0.0111 and 0.0094 mm/h, respectively. While lower growth rates were obtained on the Czapek controls, with values from 0.0092 to 0.0167 mm/h, these values appear higher than those obtained for the other fungal isolates. From estimated radial growth rates, it can be observed that 1% crude oil medium generally allows reduced growth than control Czapek medium. Similarly, it can be observed that the 5% crude oil medium allows close growth to that in 1% crude oil medium for *P. polonicum* AMF16, *P. cyclopium* AMF47, *P. olsonii* AMF72, *P. cyclopium* AMF74 and *P. mononematosum* AMF79 but reduced growth or no growth for the seven other isolates. Interestingly, several isolates belonging to the same species exhibited different behaviors in the presence of crude oil, suggesting an intraspecific variability. For example, regarding the species *Penicillium chrysogenum*, two are able to grow at different crude oil concentrations (AMF4 and AMF40) while two others are inhibited (AMF33 and AMF76). For *P. cyclopium*, one isolate was inhibited by crude oil (AMF73) while two other isolates showed an ability to grow (AMF47 and AMF74). It is also worth mentioning that two isolates of *A. versicolor* displayed reduced growth (AMF97) or no growth (AMF75) at 5% crude oil, suggesting further evidence of an intraspecific variability of biological response to crude oil. Taken together these data confirm that the crude oil can be utilized as carbon source by the 12 isolates but suggest a potential inhibitory effect of the crude oil at levels higher than 1%.

### 3.3. Crude Oil Degradation

The 12 most promising isolates were then investigated for their ability to degrade crude oil through the change in color of the culture media supplemented with 0.1% DCPIP, as detailed in Marchand et al. 2017 [120]. DCPIP is an electron acceptor that can be used as an indicator of microbial metabolism.

Initially blue in its oxidized form, this dye is reduced and discolored by electrons from the respiratory chain. Here, supplementing our media with DCPIP allowed us to determine the ability of fungal isolates to utilize crude oil as a carbon and energy source. The negative controls (Czapek medium with sucrose replaced by crude oil, supplemented with DCPIP and without any fungal inoculation) remained all blue. Among the 12 selected isolates, only five produced a visible halo, as illustrated in Figure 3. After 6 days of growth the discoloration halos of four fungal isolates were higher than those determined from the Czapek controls. Only *P. mononematosum* AMF79's degradation halo was similar to the one obtained using the Czapek control. The highest difference was obtained for *P. cyclopium* AMF47, with a halo ~1.64 times higher than when sucrose was replaced by crude oil, followed by *P. polonicum* AMF16, *P. chrysogenum* AMF40 and *P. cyclopium* AMF74 with a halo ~1.39, 1.33 and 1.25 times higher, respectively. Based on these results, it can be asserted that AMF47, AMF16, AMF40 and AMF74 are able to metabolize Arabian Light crude oil through the synthesis of specific enzymes when grown with hydrocarbons as a unique carbon source.

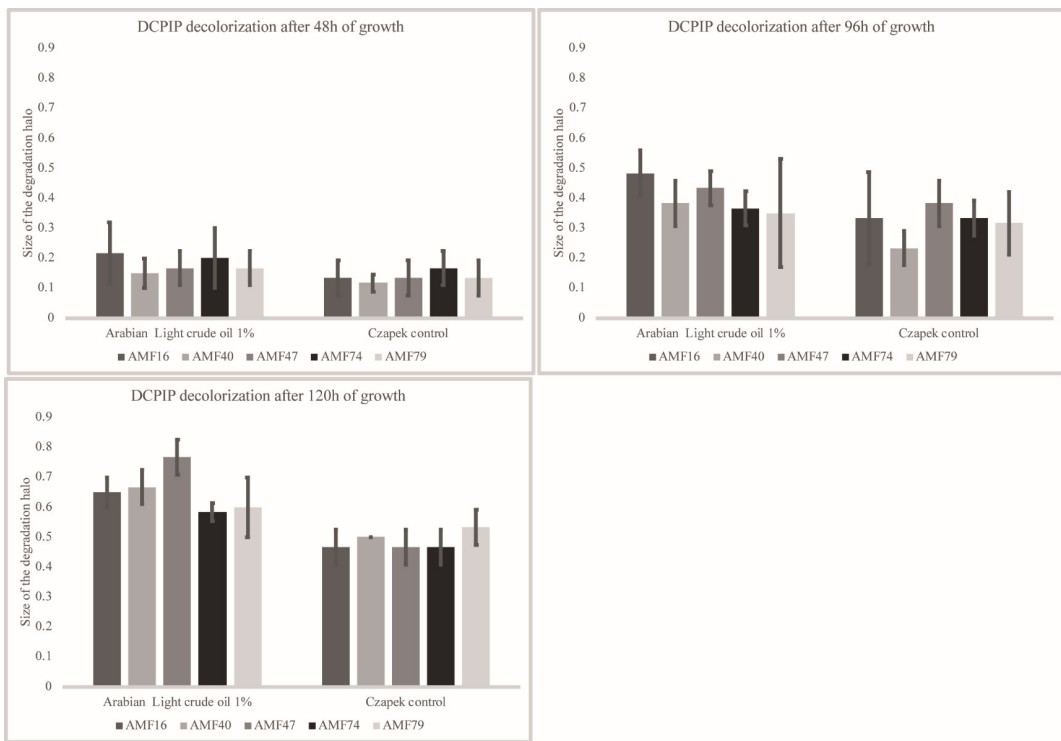

**Figure 3. Level of 2,6-dichlorophenol indophenol (DCPIP) discoloration in terms of the average size of a halo around the fungal colonies after 48, 96 and 120 hours of growth (*n* = 3).** AMF16: *Penicillium polonicum*; AMF40: *Penicillium cyclopium*; AMF47: *Penicillium cyclopium*; AMF74: *Penicillium cyclopium* and AMF79: *Penicillium mononematosum*.

*3.4. Biosurfactant Production*

Using two basic screening approaches to detect the presence of biosurfactants produced by fungi, namely the oil spreading and parafilm tests, a contrasted response was observed (Figure 4). However, according to these tests, among the 12 fungal isolates, four showed high surfactant activities as revealed by positive hits using cell-free supernatants. Interestingly, these four fungal isolates were those showing the highest growth capability using crude oil as a unique carbon source as well as those showing the highest discoloration halo based in the DCPIP-based method. They correspond to *P. polonicum* AMF16, *P. chrysogenum* AMF40 and *P. cyclopium* AMF47 and AMF74.

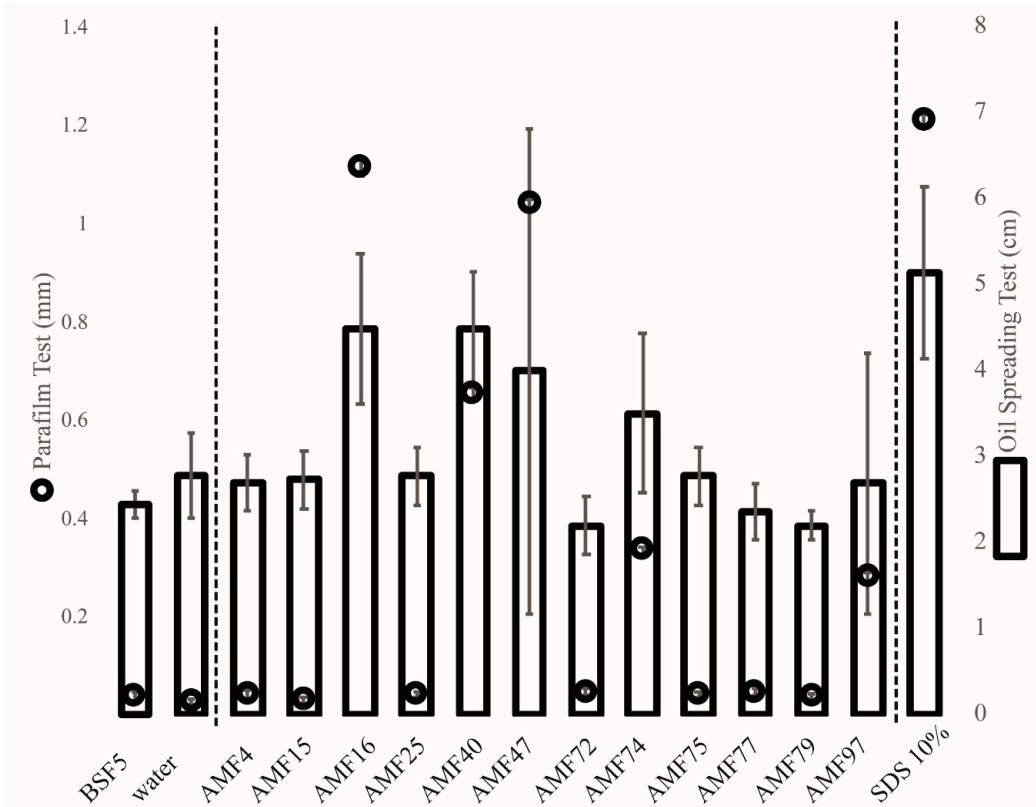

**Figure 4.** **Ability of selected fungal isolates to produce biosurfactants based on the oil spreading test (*n* = 3) and the parafilm test (*n* = 7).** AMF4: *Penicillium chrysogenum*; AMF15: *Aspergillus sydowii*; AMF16: *Penicillium polonicum*; AMF25: *Penicillium steckii*; AMF40: *Penicillium cyclopium*; AMF47: *Penicillium cyclopium*; AMF72: *Penicillium olsonii*; AMF74: *Penicillium cyclopium*; AMF75: *Aspergillus versicolor*; AMF77: *Aspergillus sydowii*; AMF79: *Penicillium mononematosum* and AMF97: *Aspergillus versicolor*.

## 4. Discussion

Millions of liters of oil enter the marine environment each year from natural and anthropogenic sources [6]. Numerous studies have investigated the biodegradation capability of crude oil by marine bacteria [32,121–125]. However, little is known about the degradation of hydrocarbons by fungi occurring in contaminated marine habitats. In this study, the analysis of seawater samples taken from the port of Oran, a chronically oil-contaminated site on the Algerian coast, allowed the isolation of 84 filamentous fungal isolates. The observed biodiversity, with 30 different taxa, demonstrates the existence of complex culturable fungal communities living in hydrocarbon-contaminated marine habitats, which appears consistent with previous studies that reported a high fungal diversity in oil-polluted marine environments [45,126].

All fungal isolates were affiliated to the Ascomycota phylum, which is the largest and most widespread taxonomic phylum in the marine environment [127,128], and more precisely to the *Penicillium*, *Aspergillus* and *Cladosporium* genera. The dominance of Ascomycota in our dataset appears consistent with a recent report from Jones et al. (2015) [85], listing a total of 1112 marine fungal species of which 805 belong to Ascomycetes. Interestingly, one study focusing on culturable marine fungal communities from shellfish farming areas along the west coast of Algeria, near the city of Oran, highlighted a high number of isolates affiliated to Ascomycetes with a large majority assigned to the *Penicillium*, *Aspergillus*, *Cladosporium* and *Trichoderma* genera [129]. The *Penicillium* and *Aspergillus* genera have already been reported as dominant in hydrocarbon-contaminated samples such as seawater and marine sediments [45,67,130], but also in numerous marine habitats in which their representatives often account for ~50% of the fungal diversity [131]. The *Cladosporium* genus is

also part of the frequently isolated genera but is generally associated with marine organisms such as cnidarians [132] and sponges [133,134]. Following the 2010 Deepwater Horizon oil spill, which released 134 million gallons of oil into the Gulf of Mexico, Bik et al. (2012) [126] analyzed the microbial eukaryotic communities through a metabarcoding approach. Their results revealed two distinct fungal community structures, one dominated by *Cladosporium* OTUs (Operational Taxonomic Units), with a close relationship to *C. cladosporioides*, and a second one dominated by *Alternaria* OTUs. In addition to these dominant OTUs, other OTUs phylogenetically related to the *Aspergillus*, *Acremonium* and *Acarospora* (Ascomycota), *Rhodocollybia* (Basidiomycota) and *Rhizopus* (Mucoromycota) genera were reported. Numerous representatives of these genera have already been shown to degrade hydrocarbon compounds [135,136] and interestingly, most of these genera (namely, *Alternaria, Cladosporium, Aspergillus* and *Acremonium*) has been found in our study.

Among the 84 fungal isolates, 30 affiliated to 11 taxa have been reported here for the first time in the marine environment. Most of these species are commonly reported in soils [137] or associated to terrestrial plants [138,139]. All other species have already been found in the marine environment, from seawater [45], marine sediments [45,64,65] or associated with various types of marine organisms, such as the marine plant *Posidonia oceanica* [78], sponges [71,73,74,133,134] or marine algae [72,80,81]. As all fungal isolates obtained here appear ubiquitous, the term "marine-derived fungi" can be used as no complementary analyses were processed to highlight any active ecological role in the marine environment [140]. These isolates may thus be of terrestrial, freshwater or marine origin.

In order to select the most promising isolates with effective hydrocarbon-degradation capabilities, each marine fungal isolate obtained was tested for its ability to grow in the presence of crude oil as a unique carbon source. Among the 84 isolates, none was able to grow in the presence of 0.1% crude oil, indicating that this level does not represent a sufficient carbon intake for growth, and 12 appear able to utilize crude oil for growth. Using a DCPIP-based approach as a proxy to highlight the ability to metabolize crude oil, four isolates, namely *P. polonicum* AMF16, *P. chrysogenum* AMF40, *P. cyclopium* AMF47 and *P. mononematosum* AMF79, led to significant discoloration of the DCPIP and thus appear truly able to metabolize hydrocarbons (in the presence of 1% of crude oil). This is consistent with a previous study focusing on the culturable mycobiota of a Mediterranean marine site after an oil spill and revealing four strains, among 142 isolates, able to degrade hydrocarbons, namely *Aspergillus terreus, Trichoderma harzianum, Penicillium citreonigrum* and one Lulworthiales sp. [45]. Using sand samples from Pensacola beach (Gulf of Mexico) contaminated with crude oil, Al-Nasrawi (2012) [141] isolated 16 strains from which four were able to degrade crude oil and were identified as *Aspergillus niger, Penicillium decumbens, Cochliobolus lutanus* and *Fusarium solani.* Simister et al. (2015) [44] obtained three fungal isolates from oil-soaked sand patties originating from the Deepwater Horizon oil spill with an ability to degrade diverse oil-derived compounds such as n-alkanes, pristine, phytane and PAHs. Using samples of cnidarians collected on the north coast of São Paulo [132], Passarini et al. (2011) [2] revealed two isolates, identified as *Aspergillus sclerotiorum* and *Mucor racemosus,* with an ability to degrade pyrene and benzo[a]pyrene. Interestingly, most of these species are ubiquitous, indicating that marine-derived fungi may represent an untapped reservoir of hydrocarbonoclasts.

In addition to the capability to utilize crude oil, the biosurfactant production potential of the 12 selected isolates was confirmed by two screening tests, i.e., the oil spreading and the drop-collapse tests. From the 12 isolates screened, four were able to produce biosurfactants (*P. polonicum* AMF16, *P. chrysogenum* AMF40 and *P. cyclopium* AMF47 and AMF74). Biosurfactant production appears common for fungi [55,142,143] and also bacteria [124,144–147] as biosurfactants may facilitate hydrocarbon bioavailability. No quantitative study was performed here to determine the concentration of biosurfactants; however, Morikawa et al. (2000) [148] reported that the oil displacement area in the oil spreading test was directly proportional to the concentration of biosurfactants in the solution. Biosurfactant production appeared well correlated to crude oil-degradation capability as the four fungal isolates able to produce high amount of biosurfactants were those that efficiently utilized crude oil.

Among the 84 isolates obtained, four appear as the most promising candidates for oil bioremediation and were all affiliated to the *Penicillium* genus. Numerous *Penicillium* species have been recorded either in the marine environment [45,74,116,149] or from polluted soils where they have been shown to degrade a wide range of recalcitrant pollutants such as PAHs, heavy metals and phenolic compounds [33,150–152]. Based on our study, three *Penicillium* species, *P. cyclopium*, *P. chrysogenum* and *P. polonicum*, appear promising for more in-depth studies in order to reveal their biotechnological interest in the field of bioremediation. If *P. cyclopium* has never been reported before in the marine environment, *P. chrysogenum* and *P. polonicium* have already been isolated from different marine habitats such as marine sediments, and seawater [45,64,65] and associated with marine sponges, marine algae [71–73,80,134] or the marine plant *Posidonia oceanica* [78]. However, their ability to degrade crude oil in contaminated marine environments has rarely been demonstrated. For example, Elshafie et al. (2007) [66] isolated and characterized one fungal isolate of *P. chrysogenum* able to degrade n-alkanes and crude oil from tar balls collected on the beaches of Oman. In another study, Velez et al. (2020) [116] obtained 25 fungal isolates from marine sediments collected in oil-drilling regions (Gulf of Mexico) with some showing an ability to degrade hexadecane and 1-hexadecene as unique carbon sources, notably those affiliated to the genera *Aureobasidium*, *Penicillium*, *Phialocephala* and *Cladosporium*. As a proof-of-concept, a transcriptomic analysis using a species affiliated to the genus *Penicillium* highlighted up-regulated genes involved in specific functions, such as the metabolism of six-carbon carbohydrates.

This study has revealed that marine fungi appear as a promising microbial resource for bioremediation of crude oil pollution and now raises complementary analyses on the most promising isolates to accurately determine the kind of hydrocarbons that are metabolized and the degradation dynamics using GC-MS-based approaches (Gas Chromatography-Mass Spetrometry). This study also highlights the importance of intraspecific variability and emphasizes the relevance of high-throughput culturing strategies to obtain large collections of microbial isolates, coupled with high-throughput screening approaches to efficiently determine the most promising isolates, i.e., those able to efficiently utilize hydrocarbons and produce biosurfactants, as candidates for bioremediation applications within the frame of bioaugmentation or biostimulation processes.

**Author Contributions:** Conceptualization, A.M.-B., M.B., and G.B.; methodology, A.M.-B., M.-E.L., S.D., M.Q. and N.N.v.L.; software, A.M., M.-E.L., S.D., M.Q. and N.N.v.L.; validation, A.M.-B., M.B., M.-E.L. and G.B.; formal analysis, A.M. and G.B.; investigation, A.M.-B., M.B. and A.M.; resources, A.M.-B., M.B. and G.B.; data curation, A.M.-B. and G.B.; writing—original draft preparation, A.M. and G.B.; writing—review and editing, all authors; visualization, A.M., N.N.v.L. and G.B.; supervision & project administration, A.M.-B., M.B., E.C. and G.B.; funding acquisition, A.M.-B., M.B. and G.B. All authors have read and agreed to the published version of the manuscript.

**Funding:** This research received no external funding except a partial support for student exchange (A.M.) by the University of Oran.

**Acknowledgments:** The authors wish to thank Jérome Lepioufle for his help in proofreading the english and correcting the manuscript. Authors also want to thank the reviewers for their valuable comments.

**Conflicts of Interest:** The authors declare no conflict of interest.

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
