# Peer review of "Highlighting the Crude Oil Bioremediation Potential of Marine Fungi Isolated from the Port of Oran (Algeria)"

_diversity, doi:10.3390/d12050196_

Round 1

Reviewer 1 Report

The article titled "Highlighting the crude oil bioremediation potential of marine fungi isolated from the port of Oran (Algeria)" addresses a very interesting and important topic. Today many researcher focus their work to find, organisms, substances and technologies able to solve environmental problems. Fungi, especially those derived from the marine environment, have enormous potential in this context.

The article is very interesting, however it has some aspects that could be improved

  1. The introduction cite many articles, however, references to works focused to study the fungal communities in the intertidal zone and occurrence of microfungion various marine substrates, such as intertidal sediments drifted woods, animals, etc. are missing.
  2. In materials and methods line 80-83 is not clear the sampling method,  which part of water was sampled… perhaps the surface? the water just below the surface?... please clarify
  3. Line 86: the plate were incubatedat 27 ° C. Why did you choose this incubation temperature? Is it a typo? Subsequent experiments were carriedout at different temperatures (28°C). Please specify or correct;
  4. Line 95. Aspergillus, Penicillium and Cladosporiumwere identifiedwith additional primers. What about the other genera? For example, BT is a good marker also for Chaetomium and Scedosporium.
  5. Lines 114-115. You mention phylogenetic trees, where are they?
  6. Line 151 check “2.5.106”typo?
  7. Line 152 “incubated at 28 °C” why?
  8. Line 167 have you submitted the sequences to Genbank, could you report the accession numbers?
  9. Line 176 “genus(” need space
  10. Line 197 no space “AMF16 ,”

  11. Line 212 “(Marchand et al. 2017)” without brackets
  12. Line 264 “Alternairai” correct
  13. Line 265 Acremoniumin italic
  14. in line 238 are mentioned"fungi occurring in contaminated marine habitats" and "fungal communities living in hydrocarbon-contaminated marine habitats" (line 242) , etc., while inline 290 you mention "marine-derived fungi" since “Diversity” is wide and multidisciplinary (not only mycologist), it would be nice clarifying and explain the concept of marine derived fungi.

Author Response

Answers to reviewers 

We thank the reviewers for their valuable comments. These comments are very constructive, and will help us to improve the manuscript, specifically in terms of clarifying our methodology.We have provided our responses to the reviewers’ comments, and corresponding changes have been made to improve the manuscript.

Responses to Reviewer1 

  1. The introduction cite many articles, however, references to works focused to study the fungal communities in the intertidal zone and occurrence of microfungi on various marine substrates, such as intertidal sediments drifted woods, animals, etc. are missing.

We thank the reviewer for this good suggestion. We have now included a new paragraph in the Introduction adressing the fungal communities in the intertidal zone on line 34. The references used have been added to the reference list.

2.In materials and methods line 80-83 is not clear the sampling method, which part of water was sampled… perhaps the surface? The water just below the surface?... please clarify

Thank you for your comment. For more accuracy, the seawater samples were collected just below the surface about 20 cm depth. The text has now been modified to provide better clarification on line 93.

  1. Line 86: The plates were incubated at 27 ° C. Why did you choose this incubation temperature? Is it a typo? Subsequent experiments were carried out at different temperatures (28°C). Please specify or correct.

Thank you for your comment. This was a typo, sorry for this mistake. All plates were incubated at 24°C except for biosurfactant production where we used a protocol based on an incubation temperature of 28°C.The correction has been made at the line 99.

4.Line 95. Aspergillus,Penicilliumand Cladosporiumwere identified with additional primers. What about the other genera? For example, BT is a good marker also for Chaetomium and Scedosporium.

Thank you for your comment. We do agree that Beta-tubulin is a good genetic marker for Chaetomium and Scedosporium. However, here using ITS sequencing, AMF_31 was 100% assigned to Chaetomium globosum and thus not requiring sequencing of complementary markers. Regarding Scedosporium, using ITS sequencing, AMF_67, AMF_71 and AMF_28 were assigned at 99% to Scedosporium dehoogii but also to other Scedosporium species. As these isolates did not show interesting degradation potential, we did not sequence any other genetic markers. However,we decided to modify the assignation of these 3 isolates to Scedosporium sp. instead of Scedosporium dehoogii (Table 1).

  1. Lines 114-115. You mention phylogenetic trees, where are they?

Sorry for this. Phylogenetic trees have been processed for our own use. We thought that a table was a better representation compared to multiple phylogenetic trees. We have deleted the sentence related to phylogenetic trees.

  1. Line 151 check “2.5.106”typo?

Line 164, we have revised the typo.

  1. Line 152 “incubated at 28 °C” why?

Already answered. We have used a specific protocol for biosurfactant production with an incubation temperature of 28°C.

  1. Line 167 have you submitted the sequences to Genbank, could you report the accession numbers?

Accession numbers have been reported. Please see end of part 2.2, Line 127.

  1. Line 176 “genus(” need space

Line 190, we have revised the typo.

  1. Line 197 no space “AMF16 ,”

Line 210, we have revised the typo.

  1. Line 212 “(Marchand et al. 2017)” without brackets

Line 225, the correction has been made

  1. Line 264 “Alternairai” correct

Line 278, we have revised the typo.

  1. Line 265 Acremoniumin italic

Line 279, the correction has been made

  1. in line 238 are mentioned "fungi occurring in contaminated marine habitats" and "fungal communities living in hydrocarbon-contaminated marine habitats" (line 242) , etc., while inline 290 you mention "marine-derived fungi" since “Diversity” is wide and multidisciplinary (not only mycologist), it would be nice clarifying and explain the concept of marine derived fungi.

Thank you for this comment. As pointed, it is important to define the term “marine-derived”. The concept of this term has thus been explained lines 285: “As all fungal isolates obtained here appear ubiquitous, the term “marine-derived fungi” can be used as no complementary analyses were processed to highlight any active ecological role in the marine environment (Pang et al. 2016). These isolates may thus be of terrestrial, freshwater or marine origin”.

Reviewer 2 Report

The article is good but minor changes must be done.

The introduction must be shorter, some paragraphs must be in the discussion.

The references list are not as requested by the journal

Please correct some errors

  1. line 151; 2.5 x 106
  2. line 262; change the word illuminated
  3. line 264; Alternaria
  4. line 265; Acremonium in italic
  5. line 303; candidates

The introduction must be shorter, some paragraphs must be in the discussion.

The references list are not as requested by the journal

Please correct some errors

  1. line 151; 2.5 x 106
  2. line 262; change the word illuminated
  3. line 264; Alternaria
  4. line 265; Acremonium in italic
  5. line 303; candidates

Author Response

Answers to reviewers 

We thank the reviewers for their valuable comments. These comments are very constructive, and will help us to improve the manuscript, specifically in terms of clarifying our methodology.We have provided our responses to the reviewers’ comments, and corresponding changes have been made to improve the manuscript.

Responses to Reviewer2:

  1. The introduction must be shorter; some paragraphs must be in the discussion.

Thank you for this comment. The introduction has been revised following reviewer 1 instructions to include on new paragraph. As underlined by reviewer 1, the journal “Diversity” is wide and multidisciplinary and many readers may not be mycologists. We therefore decided not to shorten the introduction in order to introduce this manuscript with a complete state-of-the-art on this topic.

  1. The references list is not as requested by the journal

Thank you for pointing this out. We have modified the reference list as per the journal guidelines.

  1. Line 151; 2.5 x 106

Line 164, the correction has been made.

  1. Line 262; change the word illuminated

On line 276, we have replaced in the sentence the word “illuminated” by “reported”.

  1. Line 264; Alternaria

Line 278, we have revised the typo.

  1. Line 265; Acremonium in italic

Line 279, the correction has been made.

  1. Line 303; candidates

Line 321, we have revised the typo.